# Cerebral Metabolic Rate of Glucose and Cognitive Tests in Long COVID Patients

**DOI:** 10.3390/brainsci13010023

**Published:** 2022-12-22

**Authors:** Kamilla W. Miskowiak, Johanne L. Bech, Alexander Cuculiza Henriksen, Stine Johnsen, Daria Podlekareva, Lisbeth Marner

**Affiliations:** 1Department of Psychology, University of Copenhagen, Øster Farimagsgade 2A, DK-1353 Copenhagen, Denmark; 2Neurocognition and Emotion in Affective Disorders Centre (NEAD), Psychiatric Centre Copenhagen, Copenhagen University Hospital, Rigshospitalet, Blegdamsvej 9, DK-2100 Copenhagen, Denmark; 3Department of Clinical Physiology and Nuclear Medicine, Copenhagen University Hospital Bispebjerg, Bispebjerg Bakke 23, DK-2400 Copenhagen, Denmark; 4Department of Respiratory Medicine, Copenhagen University Hospital Bispebjerg, Bispebjerg Bakke 23, DK-2400 Copenhagen, Denmark; 5Department of Clinical Medicine, University of Copenhagen, DK-1165 Copenhagen, Denmark

**Keywords:** COVID-19, cognitive impairment, brain metabolism, PET, work function, quality of life, brain fog, positron emission tomography, FDG

## Abstract

Background: Common long-term sequelae after COVID-19 include fatigue and cognitive impairment. Although symptoms interfere with daily living, the underlying pathology is largely unknown. Previous studies report relative hypometabolism in frontal, limbic and cerebellar regions suggesting focal brain involvement. We aimed to determine whether absolute hypometabolism was present and correlated to same day standardized neurocognitive testing. Methods: Fourteen patients included from a long COVID clinic had cognitive testing and quantitative dynamic [^18^F]FDG PET of the brain on the same day to correlate cognitive function to metabolic glucose rate. Results: We found no hypometabolism in frontal, limbic and cerebellar regions in cognitively impaired relative to cognitive intact patients. In contrast, the cognitive impaired patients showed higher cerebellar metabolism (*p* = 0.03), which correlated with more severe deficits in working memory and executive function (*p* = 0.03). Conclusions: Hypermetabolism in the cerebellum may reflect inefficient brain processing and play a role in cognitive impairments after COVID-19.

## 1. Introduction

In the context of the Coronavirus Disease 2019 (COVID-19), many patients report psychical and/or cognitive complaints after recovery [1]. These long-term sequelae after COVID-19 are called ‘long COVID’ and commonly involve fatigue, dyspnea, and cognitive impairment [2,3,4]. Long COVID is defined as symptoms persisting longer than 12 weeks after the acute onset of COVID-19 according to the National Institute of Health and Care Excellence (2022). Approximately 10–15% of patients are affected by long COVID, which can occur across all ages [5,6] even after a mild course of illness [6,7]. However, significant cognitive sequelae are mostly found in hospitalized or severely affected patients [7,8,9]. Interestingly, the different strains of SARS-CoV-2 do not seem to differ in symptomatology of long COVID [10]. The widespread use of vaccinations may impact future prevalence of long COVID, as indicated by the finding in a recent meta-analysis of low level of evidence for risk reduction when vaccinated before SARS-CoV-2 infection [11].

In a recent case–control study, hospitalized patients with COVID-19 were found to exhibit more cognitive impairments 6 months after discharge than demographically matched patients hospitalized for other respiratory illnesses [12]. These long-term cognitive impairments are called ‘cognitive COVID’. Across studies, a common finding is that the impairments are particularly pronounced in ‘higher order’ cognitive domains, such as executive functions, sustained attention, and verbal learning and memory [9,13,14,15]. Deficits in these cognitive domains reduce people’s ability to tackle complex cognitive challenges, concentrate on tasks over longer periods of time, and keep track of things, and thereby impede their occupational and psychosocial functioning [16,17]. In accordance with this, we found that the severity of cognitive impairment after COVID-19 was associated with more impaired work function, stress, depression and anxiety symptoms both three months [9] and one year after acute illness [18].

Several recent studies and analysis of brain 2-[^18^F]fluoro-2-deoxy-D-glucose Positron Emissions Tomography ([^18^F]FDG PET) found hypometabolism in cortical, limbic/paralimbic regimes, and in the brainstem and cerebellum, which has been proposed as a cerebral biomarker of cognitive COVID [1,5,19,20,21,22]. Specifically, brain PET-scans in long COVID patients compared to healthy controls identified lower metabolic values in the bilateral rectal/orbital gyrus, the right temporal lobe, including the amygdala and hippocampus, bilateral pons/medulla brainstem, and cerebellum from three weeks to 11 months after initial infection [1,5]. Further, hypometabolism in the cerebellum has repeatedly been found to be associated with impairments in memory, attention, and executive functioning [1,5,23]. Attention and executive deficits have also been associated with prefrontal hypometabolism [23]. In contrast, hypermetabolism has been demonstrated in the more acute phase after COVID-19 [24], which was thought to reflect ongoing brain inflammation. Notably, one [^18^F]FDG PET study was unable to replicate the metabolic brain changes even in patients with severe cognitive complaints [25].

The above PET studies were performed using short static imaging assessing changes in *relative* metabolic distribution rather than in regional *absolute* brain metabolism. Assessment of such relative changes in metabolic distributions cannot detect global changes and may be biased by areas with increased or decreased metabolism. It is also a limitation of the studies that they did not directly compare patients with and without cognitive impairments after COVID-19, which precludes insights into specific metabolic changes related to cognitive impairment per se. In the present study, we therefore aimed to assess the impact of COVID-19 on quantitative cerebral metabolic rate of glucose in individuals with and without significant cognitive COVID identified from a sample of patients referred to a long COVID clinic at Copenhagen University Hospital, Bispebjerg. We hypothesized (I) that patients with cognitive long COVID sequelae would show lower absolute metabolism in frontal or limbic cortical areas compared to cognitive intact long COVID patients (i.e., hypometabolism), (II) that previously observed cerebellar hypometabolism in long COVID patients could be replicated with absolute quantification, and (III) that the difference in regional metabolism would correlate with poorer cognitive performance.

## 2. Materials and Methods

### 2.1. Participants and Recruitment

Patients were recruited consecutively from June 2020 to December 2021 from the Long COVID clinic at Copenhagen University Hospital, Bispebjerg, which they attended either (I) as a standard follow-up assessment after hospitalization with COVID-19 in the hospital department or (II) due to referral from their general practitioner because of lingering physical symptoms, mostly respiratory problems after COVID-19. The study was part of a larger study of long COVID approved by the regional ethics committee (H-20035553) [4] and all patients gave consent after written and oral information in accordance with the ethical standards as specified in the 1964 Declaration of Helsinki.

Diagnosis of COVID-19 had been made by a positive PCR test for SARS-CoV-2 from the respiratory tract or a positive COVID-19 IgG titer. All patients at the Long COVID clinic were screened for cognitive impairment with a brief objective cognitive screening test (details below) compared to the expected performance based on their age, sex, and level of education. Inclusion criteria was a score of more than 1.5 standard deviation (SD) lower than expected (‘impaired group’) or within 0.5 SD from the expected score (‘intact group’) because we wanted to compare the clinically significantly cognitively impaired with the cognitively normal individuals. Exclusion criteria were an ‘intermediate’ cognitive performance score of 0.5–1.5 SD below expected, pre-existing neuropsychiatric disorder, inability to lie still in scanner for an hour, and insufficient Danish proficiency for testing (please refer to Appendix A for overview).

### 2.2. Procedures

As part of their standard assessment at the long COVID clinic at Bispebjerg Hospital, participants underwent objective cognitive screening (details below) and completed the Cognitive Failures Questionnaire (CFQ) [26] and the EQ-5D-5L Quality of life questionnaire (EQ5D) [27]. The Charlson Comorbidity Index (CCI) [28] was also completed. The PET study was conducted on a separate occasion, also at the Bispebjerg Hospital. On this occasion, participants completed a comprehensive neuropsychological test battery and filled out the Beck Depression Inventory-II (BDI-II) [29] before undergoing PET imaging.

### 2.3. Cognitive Test Battery

The objective cognitive screening was conducted with the brief (<20 min) Screen for Cognitive Impairment in Psychiatry Danish Version (SCIP-D) [30,31] and the Trail Making Test Part B (TMT-B) [32]. SCIP-D is a test battery with five subtests that measures four domains of cognitive functioning: Verbal learning and memory, working memory, verbal fluency, and processing speed. Danish age, education and sex adjusted norms for the SCIP-D were used [31] to assess patients’ performance. We also use age and education-based norms for the Trail Making B test, which measures executive function [31].

The neuropsychological assessment on the day of PET imaging included the Rey Auditory Verbal Learning Test (RAVLT) total learning, immediate recall, delayed recall, and recognition [33,34], Coding and Digit Span Forward from the Repeatable Battery for the Assessment of Neuropsychological Status (RBANS) [35], the Letter-Number-Sequencing (LNS) test from Wechsler’s Adult Intelligence Scale 3rd edition (WAIS-III) [36], Facial Expression Recognition Task [37], and a verbal fluency test with letters S and D tests [38]. The participants were also given the Danish Reading Test for Adults (DART) [39] that assessed the estimated premorbid verbal IQ.

### 2.4. PET Scanning

The quantitative PET examinations were performed as described previously [40] after at least four hours fasting and immediately after the neuropsychological testing. Briefly, the scans were performed in a Discovery MI PET/CT (GE Healthcare, Chicago, IL, USA) after manual injection of 200 MBq of [^18^F]FDG simultaneous with initiation of a three-part dynamic PET consisting of a 15 min recording of the descending thoracic aorta, a 40 min recording of the brain and a 4 min recording of the descending thoracic aorta. A “low dose” CTs of the thorax and the brain were performed and used for attenuation correction of the PET images, and Q. Clear, a “Bayesian penalized likelihood” algorithm (GE Healthcare, Chicago, IL, USA), was used for tomographic reconstruction with a β-value of 100. Plasma glucose was measured immediately before the scanning procedure.

### 2.5. PET Analyses

To achieve the arterial input function necessary for quantification, the images of the aorta were loaded into PMOD software (version 4.3, PMOD Technologies, Zürich, Switzerland) and an ellipsoid VOI was manually drawn in the lumen of the descending aorta on an early summed image in each patient as previously described [40]. Whole blood activity concentration was extracted from the dynamic unsmoothed early images of aorta as well as the late aorta image and fitted with a triexponential model after the peak to estimate the missing plasma values during the brain part of the scan and corrected with the previously found plasma–whole blood ratio.

Cerebral Metabolic Rate of glucose (CMR_glc_) was calculated from a Gjedde–Patlak plot with plasma glucose concentration as measured, a lumped constant of 0.65, and blood volume, V_b_ fixed to 0.05. To delineate the brain volumes of interest, the PMOD Neuro Tool’s workflow for PET-only studies was used and automatically delineated by a probabilistic atlas (The AAL Neuro Atlas) [41]. For a frontal region accountable for executive functions, we chose right and left superior frontal cortex. The right and left temporal pole, hippocampus and amygdala were used as limbic structures previously shown to be involved in long COVID, and cerebellum_8 was used as representative for cerebellum also previously shown to be involved in long COVID.

### 2.6. Statistical Analyses

The statistical analyses were conducted using IBM Statistical Package for Social Sciences (SPSS) statistics 25 for windows (IBM Corporation, Armonk, NY, USA) and the statistical significance was set to an alpha-level of *p* < 0.05 (two-tailed). To examine whether data were normally distributed, the Shapiro–Wilk’s test was used. The age, sex, level of education, BMI, hospitalization, days from infection to PET-scan, and days between tests along with results from the questionnaires, the SCIP-D and the neurological test battery were compared with independent samples *t*-tests.

Expected cognitive SCIP scores were calculated based on participants’ age, sex and education levels using regression-based formula established within a pre-established normative data set [31]. Expected Trail Making Test B scores were established based on age and level of education. Cognitive domains for the neuropsychological tests were calculated by z-transforming scores from the individual neuropsychological tests based on the mean and SD of scores of 100 age-matched healthy control participants collected as part of another study [42] and creating a mean of these z-scores for each domain. RAVLT subtests were included in the verbal learning and memory domain; The LNS and verbal fluency tests were included in the working memory and executive function domain. The RBANS Coding test represented processing speed while the RBANS Digit Span Forward constituted attention. Associations between the cognitive domain scores and brain metabolism were examined with Pearson’s correlations. A global cognition composite was generated by adding together the cognitive domain scores and dividing it with the number of domains.

Hypothesis (I) and (II) regarding whether patients with cognitive impairments show hypometabolism in frontal or limbic cortical areas and in the cerebellum compared to cognitively intact patients were investigated using independent samples *t*-tests. Hypothesis (III) regarding whether lower regional metabolism correlates with poorer cognition was investigated with Pearson’s or Spearman’s correlations for normally and non-normally distributed data, respectively.

## 3. Results

### Participant Characteristics

Fourteen patients were included in the study, consisting of an “impaired” group (*n* = 8) and “intact” group (*n* = 6) based on their scores on the SCIP-D (Table 1). The impaired and intact groups were well balanced for sex (*p* = 0.65), age (*p* = 0.82), education (*p* = 0.10), days between tests (*p* = 0.11), hospitalization for COVID-19 (*p* = 0.16), depression symptoms (*p* = 0.38), quality of life (*p* = 0.63), self-reported cognitive difficulties (*p* = 0.38) estimated verbal IQ (*p* = 0.95), BMI (*p* = 0.55) and Charlson Index (*p* = 0.60) (see Table 1). There was a non-significant trend towards longer time since COVID-19 in the cognitively intact than in the cognitively impaired groups (*p* = 0.09).

Table 2 displays brain metabolic values from the two groups. Examination of extracted brain metabolism values from the regions of interest showed *higher* cerebellar metabolism in cognitively impaired compared to cognitively intact patients (t = 2.39, df = 12, *p* = 0.03). Post hoc calculation of the odds ratio revealed that for every increase in cerebellar metabolism of 1 point, the odds of belonging to the cognitively impaired category doubled (Exp(β) = 2.03). There were also non-significant trends towards higher metabolism in the cognitively impaired versus intact groups in bilateral hippocampi (t = 1.86, df = 12, *p* = 0.09) and the superior temporal pole (t = 1.79, df = 12, *p* = 0.10). There were no other statistically significant differences between groups in any other regions, although numerically the same pattern of higher metabolism was seen in cognitively impaired patients (*p*-values > 0.12). Post hoc comparison of brain metabolism between hospitalized and non-hospitalized patients revealed no significant differences between groups across these regions (*p*-values > 0.10).

Comprehensive neuropsychological testing revealed impairment of working memory and executive function (t = −2.58, df = 8.93, *p* = 0.03) in the impaired versus the intact groups with a large effect size (Table 3 and Table 4). In contrast, the other cognitive domains where comparable between groups (*p*-values > 0.25). Notably, deficits in working memory and executive function correlated with higher cerebellar metabolism (*p* = 0.03), as identified in the impaired group, and with higher metabolism in the bilateral superior temporal pole (*p* = 0.03), amygdala (*p* = 0.01), thalamus (*p* = 0.04) and vermis (*p* = 0.04).

## 4. Discussion

In this PET imaging study, we compared absolute measures of brain metabolism in 14 patients from a long COVID clinic pre-screened to have either clinically relevant cognitive impairments (*n* = 8) or no cognitive impairment (*n* = 6). We did not replicate previous findings of hypometabolism in frontal, limbic, and cerebellar regions [1]. In contrast with the hypothesis, we found *hyper*metabolism in the *cognitively impaired* patients, and this correlated with poorer cognitive performance.

Patients in the impaired group exhibited particular deficits in working memory and executive function. Across all patients, the severity of working memory and executive function deficits correlated with higher metabolism in the cerebellum as well as in additional temporal and limbic regions. Importantly, the differences between groups in cerebellar metabolism and cognitive function occurred in the absence of differences in demographic or clinical characteristics.

We had expected the impaired executive functioning and working memory in the ‘cognitively impaired’ group to originate from aberrant metabolism in the prefrontal cortex that is critical for higher order cognitive functions. The observation of specific hypermetabolism in the cerebellum was therefore unexpected. Nevertheless, the cerebellum was recently found to be involved in multiple cognitive domains and affective processes [43,44,45]. This broad supportive role of the cerebellum across cognitive domains may explain why we found cerebellar hypermetabolism to be a key correlate of cognitive impairment. The cerebellar hypermetabolism may represent excessive recruitment of this region in cognitive processing in the impaired group and thus be a biomarker of inefficient brain processes. This interpretation is consistent with evidence from functional magnetic resonance imaging (fMRI) studies for task-related neuronal *hyper*activity being a marker of inefficient brain functioning [46]. In line with this, cerebellar hypermetabolism correlated with the severity of cognitive impairment in our sample. However, the cerebellar hypermetabolism in cognitively impaired versus intact post-COVID-19 patients contrasts with prior observations of *hypo*metabolism in the cerebellum and limbic regions after COVID-19 [1,5,19,20,21,22]. Methodological differences may explain this discrepancy. Specifically, we compared patients with ‘clinically significant’ cognitive impairments to patients who were ‘cognitively intact’. In contrast, previous studies simply compared individuals with and without previous COVID-19 [1,5,24]. An advantage of our design was that it enabled *direct* insight into the neuronal underpinnings of cognitive sequelae of COVID-19 in the absence of confounding variables because the only difference between groups was their cognitive status. Another possible explanation for the discrepant findings was that we investigated *absolute* rather than *relative* regional brain metabolism of glucose, of which the latter could be biased by areas with increased or decreased metabolism.

We propose that the reason for the cerebellar hypermetabolism and deficits in working memory and executive function may be exhaustion and fatigue (‘brain fog’), which is often reported by patients with cognitive COVID. A possible biological cause for this fatigue is systemic dysregulation of the immune system, as reflected by heightened inflammatory response after COVID-19. Indeed, elevation of proinflammatory markers, such as cytokines and D-dimers, has been associated with poorer cognitive function after COVID-19 [3,9,47]. In keeping with this, analysis of post-mortem brain samples from patients who died with severe COVID-19 revealed cortical changes that mirror those seen in old age, including activation of genes associated with inflammation and stress and deactivation of cognition- and plasticity-relevant genes [48]. Based on this evidence, treatments with anti-inflammatory actions have gained great research interest for targeting cognitive sequelae of COVID-19 [49,50,51] While findings from intervention studies are so far inconclusive [49], there is a need for further larger scale randomized controlled trials investigating the potential cognitive benefits of anti-inflammatory treatments in patients with cognitive COVID.

The main strength of the study is the rigorous cognitive assessment and the quantitative PET glucose measurements performed on the same day, allowing for direct comparison of cognition and glucose consumption and the possibility to evaluate global cerebral effects. The main limitation of our study is the sample size (*n* = 14), which limits the statistical power of the study and could have led to type 2 errors. Another limitation is that the representativeness of the sample is limited to patients who have either been hospitalized or have been referred to a long COVID clinic by their GP due to long COVID symptoms. Therefore, the findings do not represent sequelae of COVID-19 in the general population. However, referrals from GPs were based primarily on persistent *respiratory* symptoms rather than cognitive complaints, which indicates that we do not have a selection bias for cognitive impairment in our participants. Finally, it is a limitation that we did not have patients’ pre-COVID cognitive measures. Nevertheless, the SCIP-D accommodates for this to some extent by adjusting expected scores according to the individual patient’s age and education levels. This enables individualized age- and education-based norms, which provide a good estimation of patients’ pre-COVID function.

In conclusion, we did not replicate previous findings of hypometabolism in limbic and cerebellar regions. Unexpectedly, we found hypermetabolism in the cerebellum as a key neural correlate underlying impairment in working memory and executive function after COVID-19. Based on this preliminary evidence, larger PET imaging studies comparing patients with and without cognitive COVID are now warranted to further investigate the role of cerebellar hypermetabolism in patients’ impaired executive functions and its association with demographic and clinical variables and blood-based biomarkers of inflammation.

## Figures and Tables

**Table 1 brainsci-13-00023-t001:** Demographic and clinical characteristics of the cognitively impaired and intact.

	Impaired (*n* = 8)	Intact (*n* = 6)	*p*-Value
Age (mean ± SD)	54 ± 15	56 ± 13	0.82
Sex (Women; *n* (%))	5 (63%)	3 (50%)	0.67
Education (mean ± SD)	13 ± 3	17 ± 3	0.10
Estimated verbal IQ (DART, mean ± SD)	112 ± 2	112 ± 8	0.95
Days between tests (mean ± SD)	41 ± 26	63 ± 17	0.11
Days from infection to PET-scan (mean ± SD)	210 ± 35	329 ± 152	0.09
Hospitalization (Yes; *n* (%))	4 (57%)	1 (17%)	0.16
Depression rating (BDI, mean ± SD)	14 ± 7	11 ± 6	0.38
Quality of life (EQ-5D, mean ± SD)	5 ± 4	6 ± 4	0.63
BMI (mean ± SD)	27 ± 4	26 ± 3	0.55
Self-reported Cognitive Distortions (CFQ, mean ± SD)	37 ± 10	44 ± 15	0.38
Comorbid conditions (CCI, mean ± SD)	2.6 ± 1.7	3.8 ± 4.3	0.60

**Table 2 brainsci-13-00023-t002:** Cerebral Metabolic rate of glucose (μmol 100 g^−1^ min^−1^) in the regions of interest in cognitively impaired and intact patients.

	Impaired (*n* = 8)	Intact (*n* = 6)	*p*-Value
Hippocampus (mean ± SD)	16.44 ± 2.65	14.08 ± 1.83	0.09
Amygdala (mean ± SD)	17.19 ± 3.31	14.58 ± 1.43	0.10
Thalamus (mean ± SD)	27.44 ± 5.99	23.17 ± 4.05	0.16
Superior Frontal Gyrus (mean ± SD)	32.25 ± 7.09	28.67 ± 4.65	0.31
Superior Temporal Pole (mean ± SD)	20.19 ± 3.90	16.92 ± 2.52	0.10
Gyrus Rectus (mean ± SD)	28.13 ± 5.62	24.67 ± 3.01	0.19
Pons (mean ± SD)	17.63 ± 2.67	16.33 ± 5.39	0.56
Cerebellum (mean ± SD)	22.88 ± 4.45	18.33 ± 1.37	0.03 *
Vermis (mean ± SD)	24.25 ± 4.53	20.67 ± 3.14	0.12

** p* < 0.05.

**Table 3 brainsci-13-00023-t003:** Cognitive test results in cognitively impaired, intact and Norm (SCIP) based on age, education and sex norms [29].

	Impaired (*n* = 8)	Intact (*n* = 6)	Norm (SCIP)	*p*-Value Impaired vs. Intact Intact vs. Norm Impaired vs. Norm All COVID vs. Norm
SCIP total score (mean ± SD)	62 ± 6	79 ± 5	76 ± 4	0.001
				0.30
				0.001
				0.06
SCIP 1 (Verbal Learning, mean ± SD)	17.8 ± 3.3	22.7 ± 1.8	22.4 ± 1.1	0.009
				0.66
				0.001
				0.06
SCIP 2 (Working Memory, mean ± SD)	16.7 ± 5.0	21.0 ± 1.4	19.9 ± 0.7	0.07
				0.04
				0.04
				0.39
SCIP 3 (Verbal Fluency, mean ± SD)	10.2 ± 3.7	17.3 ± 2.3	16.3 ± 1.2	0.002
				0.24
				0.001
				0.08
SCIP 4 (Verbal Recall, mean ± SD)	7.0 ± 1.4	7.5 ± 2.1	7.2 ± 0.7	0.64
				0.64
				0.70
				0.91
SCIP 5 (Psychomotor Speed, mean ± SD)	10.0 ± 2.1	10.0 ± 1.7	10.4 ± 1.2	1
				0.58
				0.31
				0.54
Trail Making B score (s) (mean ± SD)	113 ± 25	89 ± 31	79 ± 16	0.16
				0.40
				0.003 0.04

**Table 4 brainsci-13-00023-t004:** Cognitive test results based on neuropsychological tests.

Cognitive Domains Based on Neuropsychological Test Battery (Mean ± SD)	Impaired (*n* = 8)	Intact (*n* = 6)	*p*-Value
Global composite (mean ± SD)	−0.72 ± 0.57	−0.39 ± 0.29	0.22
Working memory and executive function (mean ± SD)	−0.72 ± 0.61	−0.04 ± 0.30	0.03 *
Verbal learning memory (mean ± SD)	0.07 ± 0.75	0.26 ± 0.54	0.67
Attention (mean ± SD)	−0.58 ± 0.80	−0.58 ± 1.31	1
Psychomotor speed (mean ± SD)	−2.11 ± 1.56	−1.18 ± 1.17	0.25
Facial Expression Recognition (mean ± SD)	−0.07 ± 0.68	0.12 ± 0.45	0.64

** p* < 0.05.

## Data Availability

Due to national and local legal requirements regarding privacy issues, it is not possible to make data openly available, and we support our legislation for ethical reasons. We want to contribute relevant knowledge from our study to the greatest possible extent. Should individual data be necessary, it is possible to enter into a process for creating a data sharing agreement.

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
