# Peer review of "Cerebral Metabolic Rate of Glucose and Cognitive Tests in Long COVID Patients"

_brainsci, 2022, doi:10.3390/brainsci13010023_

Round 1

Reviewer 1 Report

Comments and Suggestions for Authors

It is a very well prepared manuscript, methodologically well described, so I consider that in the results section the results described in Tables 3 and 4 can be described a little more thoroughly with respect to the significances and their interpretation. the order of the tables should be adjusted in the manuscript as they are placed in the discussion section.

Author Response

Reviewer 1

It is a very well prepared manuscript, methodologically well described, so I consider that in the results section the results described in Tables 3 and 4 can be described a little more thoroughly with respect to the significances and their interpretation. the order of the tables should be adjusted in the manuscript as they are placed in the discussion section.

Thank you for your kind words. We have now described the results in Table 3 and 4 more thoroughly and we have moved the Tables to page 3, 5, 6 and 7. Please find attached the revised manuscript with all changes marked with red. 

Reviewer 2 Report

Comments and Suggestions for Authors

I read with great interest the paper of Miskowiak and colleagues which explored the correlation between cerebral metabolic rate and cognitive decline in Long COVID patients. This is an important piece of information since cognitive dysfunction is one of the most common symptoms associated with Long COVID. However,  I would like the authors to revise their manuscript following the aforementioned comments and suggestions:

1. I feel that the introduction is not well-elaborated. Kindly highlight that there is not much difference in Long COVID symptomatology in different SARS-CoV-2 variants and that vaccination may have protective effects against Long COVID. DOI: 10.1016/j.eclinm.2022.101624; DOI: 10.3390/v14122629

2. Please provide a detailed schematic diagram from patient recruitment up until the end of the study. Kindly elaborate as well the inclusion and exclusion criteria used for selecting participants.

3. Are the participants here age- and sex-matched?

4. Please include the strength and limitations of this study, especially that it yielded results that are contradictory to previously published studies.

5. In the discussion, can you propose a working hypothesis or mechanism for cerebellar hypermetabolism? How could this explain cognitive decline that is always associated with Long COVID.

6. How did you assure that the cognitive decline is not induced by hospitalization since 57% of those with impaired cognitive function were hospitalized?

7. Aside from calculating for the p-values, can you also include odds ratio calculation.

Author Response

Reviewer 2

I read with great interest the paper of Miskowiak and colleagues which explored the correlation between cerebral metabolic rate and cognitive decline in Long COVID patients. This is an important piece of information since cognitive dysfunction is one of the most common symptoms associated with Long COVID. However, I would like the authors to revise their manuscript following the aforementioned comments and suggestions:

  1. I feel that the introduction is not well-elaborated. Kindly highlight that there is not much difference in Long COVID symptomatology in different SARS-CoV-2 variants and that vaccination may have protective effects against Long COVID. DOI: 10.1016/j.eclinm.2022.101624; DOI: 10.3390/v14122629

Thank you for highlighting this. We have now added to the introduction:

Interestingly, the different strains of SARS-CoV-2 do not seem to differ in symptomatology of long COVID [1]. The widespread use of vaccinations may impact future prevalence of long COVID, as indicated by the finding in a recent meta-analysis of low level of evidence for risk reduction when vaccinated before SARS-CoV-2 infection [2].

  1. Please provide a detailed schematic diagram from patient recruitment up until the end of the study. Kindly elaborate as well the inclusion and exclusion criteria used for selecting participants.

This is a good point. We have now changed the description of inclusion and added a supplemental figure that show the recruitment in schematic form:

Diagnosis of COVID-19 had been made by a positive PCR test for SARS-CoV-2 from the respiratory tract or a positive COVID-19 IgG titre. All patients at the Long-COVID clinic were screened for cognitive impairment with a brief objective cognitive screening test (for details see below) compared to the expected performance based on their age, gender, and level of education. Inclusion criteria was a score of more than 1.5 standard deviation (SD) lower than expected (‘impaired group’) or within 0.5 SD from the expected score (‘intact group’) because we wanted to compare the clinically significantly cognitively impaired with the cognitively normal individuals. Exclusion criteria were an ‘intermediate’ cognitive performance score of 0.5-1.5 SD below expected, pre-existing neuropsychiatric disorder, inability to lie still in scanner for an hour, and insufficient Danish proficiency for testing (please refer to Supplemental Figure for overview).       

Based on participants’ performance on the cognitive screening test, a total of 14 participants were offered participation in the PET study on a consecutive basis if they scored either (i) at least 1.5 standard deviation (SD) lower than expected based on their age, gender, and level of education (‘impaired group’) or (ii) within 0.5 SD from the expected score (‘intact group’). An exclusion criterion was pre-existing neuropsychiatric disorder.

  1. Are the participants here age- and sex-matched?

Yes, the cognitively impaired and intact participants did not show any differences regarding age and sex, although this was not part of the inclusion criteria. This information can be found in Table 1.

  1. Please include the strength and limitations of this study, especially that it yielded results that are contradictory to previously published studies.

We agree with the reviewer that it is important to discuss the possible reasons for the contradictory results. We did not include this discussion in the limitations section but in a section before this (line 259-270):

“However, the cerebellar hypermetabolism in cognitively impaired versus intact post-COVID-19 patients contrasts with prior observations of hypometabolism in the cerebellum and limbic regions after COVID-19 [1, 5, 19-22]. Methodological differences may explain this discrepancy. Specifically, we compared patients with ‘clinically significant’ cognitive impairments to patients who were ‘cognitively intact’. In contrast, previous studies simply compared individuals with and without previous COVID-19 [1, 5, 24]. An advantage of our design was that it enabled direct insight into the neuronal underpinnings of cognitive sequelae of COVID-19 in the absence of confounding variables because the only difference between groups was their cognitive status. Another possible explanation for the discrepant findings was that we investigated absolute rather than relative regional brain metabolism of glucose, of which the latter could be biased by areas with increased or decreased metabolism.”

We also agree with the reviewer that strengths and limitations should be discussed. We have included strengths and limitations section (lines 282-296). In response to the reviewer’s request we have now made some minor changes in this section to improve clarity for the reader:

“The main strength of the study is the rigorous cognitive assessment and the quantitative PET glucose measurements performed on the same day, allowing for direct comparison of cognition and glucose consumption and the possibility to evaluate global cerebral effects. The main limitation of our study is the sample size (n=14), which limits the statistical power of the study and could have led to type 2 errors. Another limitation is that the representativeness of the sample is limited to patients who have either been hospitalized or have been referred to a long-COVID clinic by their GP due to long COVID symptoms. Therefore, the findings do not represent sequalae of COVID-19 in the general population. However, referrals from GPs were based primarily on persistent respiratory symptoms rather than cognitive complaints, which indicates that we do not have a selection bias for cognitive impairment in our participants. Finally, it is a limitation that we did not have patients’ pre-COVID cognitive measures. Nevertheless, the SCIP-D accommodates for this to some extent by adjusting expected scores according to the individual patient’s age and education levels. This enables individualised age- and education-based norms, which provide a good estimation of patients’ pre-COVID function.“

  1. In the discussion, can you propose a working hypothesis or mechanism for cerebellar hypermetabolism? How could this explain cognitive decline that is always associated with Long COVID.

We thank the reviewer for this important question. It is not easy to propose a working hypothesis based on the current data. However, we did speculate in the discussion section that the cerebellar hypermetabolism may reflect inefficient brain processing, in line with evidence from fMRI studies that hyperactivity in task-relevant regions during cognitive processing indicates a need for greater recruitment of neural resources to maintain cognitive performance levels (i.e., inefficient neural processing). Regarding possible biological mechanisms, we also speculate that such inefficient brain processing may be caused by ongoing immune dysregulation and activation of ageing-related genes after COVID-19, which we have now elaborated on a bit more (lines 271-281):

“We propose that the reason for the cerebellar hypermetabolism and deficits in working memory and executive function may be exhaustion and fatigue (‘brain fog’), which is often reported by patients with cognitive COVID. A possible biological cause for this fatigue is systemic dysregulation of the immune system, as reflected by heightened inflammatory response after COVID-19. Indeed, elevation of proinflammatory markers, such as cytokines and D-dimers, has been associated with poorer cognitive function after COVID-19 [3, 9, 47]. In keeping with this, analysis of post-mortem brain samples from patients who died with severe COVID-19 revealed cortical changes that mirror those seen in old age, including activation of genes associated with inflammation and stress and deactivation of cognition- and plasticity-relevant genes [48].”

  1. How did you assure that the cognitive decline is not induced by hospitalization since 57% of those with impaired cognitive function were hospitalized?

This is a good point. We have compared the cognitively impaired and cognitively normal patients for demographic and clinical variables including hospitalization status and found no significant differences between the groups, as specified in the results section, lines 203-209:

“The impaired and intact groups were well balanced for sex (p=0.65), age (p=0.82), education (p=0.10), days between tests (p=0.11), hospitalization for COVID-19 (p=0.16), depression symptoms (p=0.38), quality of life (p=0.63), self-reported cognitive difficulties (p=0.38) estimated verbal IQ (p=0.95), BMI (p=0.55) and Charlson Index (p=0.60) (see table 1).”

In response to the reviewer’s point, we have now also included direct comparisons of brain metabolism in hospitalized versus non-hospitalized patients (lines 219-221):

“Post hoc comparison of brain metabolism between hospitalized and non-hospitalized patients revealed no significant differences between groups across these regions (p-values>0.10).”

  1. Aside from calculating for the p-values, can you also include odds ratio calculation.

We appreciate this comment by the reviewer and have tried to integrate this in the manuscript. The calculation of the odds ratio (OR) is usually used to get a measure of association between a variable and an event occurring. In the current study, a meaningful way to provide an OR would thus be to investigate the association between metabolism in the cerebellum (which differed between groups) and the odds of belonging to the cognitively impaired group. We have now included this in the results section (lines 214-217):

“Post hoc calculation of the odds ratio revealed that for every increase in cerebellar metabolism of 1 point, the odds of belonging to the cognitively impaired category doubled (Exp(β)= 2.03).”
